# Ultrasonic super-oscillation wave-packets with an acoustic meta-lens

Ya-Xi Shen[1,5], Yu-Gui Peng [1,2,5], Feiyan Cai[3], Kun Huang[4], De-Gang Zhao[1], Cheng-Wei Qiu [2], Hairong Zheng[3] & Xue-Feng Zhu[1]

The Schrödinger equation is a fundamental equation to describe the wave function of a quantum-mechanical system. The similar forms between the Schrödinger equation and the paraxial wave equation allow a paradigm shift from the quantum mechanics to classical fields, opening up a plethora of interesting phenomena including the optical super-oscillatory behavior. Here, we propose an ultrasonic meta-lens for generating super-oscillation acoustic wave-packets with different spatial momenta and then superimposing them to a diffraction-limit-broken spot, visually represented by the ring-shaped trapping of tiny particles. Moreover, based on the focused super-oscillation packets, we experimentally verify proof-of-concept super-resolution ultrasound imaging, opening up the arena of super-oscillation ultrasonics for advanced acoustic imaging, biomedical applications, and versatile far-field ultrasound control.

[1] School of Physics and Innovation Institute, Huazhong University of Science and Technology, 430074 Wuhan, China. [2] Department of Electrical and Computer Engineering, National University of Singapore, Singapore 117583, Republic of Singapore. [3] Paul C. Lauterbur Research Center for Biomedical Imaging, Institute of Biomedical and Health Engineering, Shenzhen Institutes of Advanced Technology, Chinese Academy of Science, 518055 Shenzhen, China. [4] Department of Optics and Optical Engineering, University of Science and Technology of China, 230026 Hefei, Anhui, China. [5] These authors contributed equally: Ya-Xi Shen, Yu-Gui Peng. Correspondence and requests for materials should be addressed to F.C. (email: fy.cai@siat.ac.cn) or to C.-W.Q. (email: eleqc@nus.edu.sg) or to H.Z. (email: hr.zheng@siat.ac.cn) or to X.-F.Z. (email: xfzhu@hust.edu.cn)

Wave functions that contain frequencies below a maximum limit are ubiquitously seen in various textbooks from theoretical physics to the applied sciences. Intuitively, one may expect that a frequency-limited wave function can vary no faster than its highest frequency component. However, in 1990s, Aharonov, Berry, and others[1–8] gave counter examples to show the existence of some wave functions temporarily oscillating at the frequencies much larger than the highest Fourier component, which they termed super-oscillatory functions. After that, super-oscillatory behaviors were introduced into classical wave systems, which gives rise to various counter-intuitive effects, such as far-field super-resolution focusing[9–19]. Over the past decades, breaking the diffraction limit is a long-sought goal in optics and acoustics. Ever since the proposals of negative-refractive-indexed[20–23] and hyperbolic-refractive-indexed metamaterials[24–27], it was unveiled that the evanescent waves carrying super-resolution information will be recovered in those novel materials. However, the super-resolution focusing and imaging were merely implemented in the near field due to non-trivial medium losses[20–27]. Hence, it is of great value to focus light or sound into a deep-subwavelength spot in the far-field.

In this study, we utilize super-oscillation wave-packets to beat the diffraction limit in the far-field for ultrasound waves. We directly construct acoustic super-oscillatory functions with the time-periodic property from the wave equation[28–30]. Then, we show the approach to combine different spatial frequencies of ultrasound waves for producing a static spatial distribution of super-oscillatory wave functions via an optimization-free method. In the demonstration of ultrasound super-oscillation, we design a meta-lens of the thickness $\sim 0.13\lambda$ to project the required spatial frequency components that generate a diffraction-limit-broken spot in the far-field focal region. The created super-oscillating packet is featured with a relatively weak amplitude but a distinct pressure gradient, which ensures a non-trivial acoustic radiation force. Such force effect is experimentally observed from the ring-shaped trapping of micro-particles, which shows the spot profile. Moreover, based on the focused super-oscillation packets, we experimentally demonstrate super-resolution ultrasound imaging of subwavelength objects.

## Results

**Construction of super-oscillation packets.** We briefly introduce the construction of super-oscillation packets in the gauge of wave equation[31]. If we constrict ourselves on the one-dimensional case in the homogeneous media, where the wave function $\psi(x, t)$ satisfies

$$\frac{\partial^2}{\partial x^2}\psi(x, t) - \frac{1}{c^2}\frac{\partial^2}{\partial t^2}\psi(x, t) = 0 \tag{1}$$

where $c$ is the wave velocity. The plane-wave solution of Eq. (1) is

$$\psi(x, t) = \phi_0 e^{i(kx - \omega t)} \tag{2}$$

where $\omega$ is the angular frequency, $k$ is the wavenumber and $\omega = ck$. To construct the super-oscillation packet, we introduce a wave function comprising several plane waves with weighted coefficients

$$f(x, t) = \sum_{n=1}^{N} C_n e^{i(k_n x - \omega_n t)} \tag{3}$$

where $C_n$ is the weighted coefficient and $N$ is the total number of plane waves. Mathematically, it is efficient to reconstruct a super-oscillatory function based on the optimization-free method[32]. For example, we assume that the wave function comprises $N(=7)$ plane waves with their frequencies ranging from 1 to 5 with an equal interval of 2/3. Without loss of

generality, we define a super-oscillatory function at $t = 1.32\pi$, for which the wave function has some characteristic values $f = [0, -1, 0, 1, 0, -1, 0]$ at the fixed locations $x = [-0.9, -0.7, -0.35, 0, 0.35, 0.7, 0.9]$. Substituting all the parameters into Eq. (3), we will have seven unknown parameters $C_n = 1, 2, \cdots, 7$. Utilizing the inverse of linear matrix equation, we obtain the solution of $C_n = [-8647.319939 - 13625.9959i, -118908.3682 - 86391.98649i, -497990.3477 - 127862.1894i, -904681.5947 + 114287.8633i, -769934.2003 + 423274.9082i, 280547.6819 + 339123.8915i, -27837.78904 + 85675.90495i]$, which are the raw data without normalization.

Figure 1 shows the evolving behavior of the constructed wave function at different times. In Fig. 1a, the wave function is a band-limited function at the initial time $t = 0$. The super-oscillation occurs as the time evolves near $t = 1.32\pi + 3\pi M$, where $M$ is an integer and $3\pi$ is its period. The periodicity is clearly displayed for one-dimensional case in Fig. 1b. Figure 1c presents the evolving details at the designed super-oscillation time zone around $t = 1.32\pi$, where a very fast oscillation occurs near $x = 0$. To quantitatively evaluate the constructed super-oscillation, we plot the real and imaginary parts of wave functions in Fig. 1d. Our results show that both parts of the wave function oscillate faster than its maximum frequency. In Supplementary Video 1, we show the dynamic evolution of the constructed wave function in the time range of $t = 0 \sim 1.39\pi$.

**Super-oscillation packets in an acoustic meta-lens.** In order to demonstrate acoustic super-oscillation, we design an optimization-free meta-lens that establishes a mapping between the time frequencies of super-oscillatory function and the spatial frequencies of ultrasound. Figure 2a shows the fabricated sample. For a single-belt Fresnel zone plate, as shown in Fig. 2b, the diffraction intensity at the focal region is $I = C_n |J_0(kr \sin \alpha_n)|^2$, where the first minimum is $r = 0.38\lambda / \sin \alpha_n$[32]. Since $\sin \alpha_n \leq 1$, the diffraction limit is thus $r_D = 0.38\lambda$. To break the diffraction limit in the confined area $[-\lambda/2, \lambda/2]$, as schematically shown in Fig. 2c, we need to utilize super-oscillation. We define the normalized pressure field amplitude on the target plane $A_n(r) = p_n(r)/C_n$ with $C_n = p_n(0)$, and the spatial frequency $\eta_n = \sin \alpha_n / \lambda$. The diffraction field induced by the meta-lens is

$$p(r) = \sum_{n=1}^{N} C_n A_n(r) \tag{4}$$

where the meta-lens projects $N$ different spatial frequency components. We preset the super-oscillatory function that satisfies $p(l_m) = p_m$ at the positions $r = \{l_m\}_{m=1}^{M}$ on the target plane. Then, by substituting all the parameters into Eq. (4), we will obtain the coefficients $C_n$. Here, an explicit super-oscillatory focal spot is designed with the radius of the principal lobe $0.3\lambda$ beyond the diffraction limit ($<0.38\lambda$), on the plane $z = 5.2\lambda$. In Fig. 2d, the pseudo-colored curves show the normalized ultrasound intensity distributions from different spatial frequency components $\eta_n$, while the black curve synthesized from the delicate combination of all components is featured with super-oscillation.

Figures 3a, b display the simulated and measured intensity profiles on the $x$-$z$ plane, respectively. The result shows that a super-oscillatory field exists at $z = 5.2\lambda$. Figures 3c, d present the simulated and measured intensity profiles in the field of view ($2\lambda \times 2\lambda$) on the $x$-$y$ plane at $z = 5.2\lambda$, respectively. Outside the field of view on the $x$-$y$ plane, the acoustic energy accounts for $\sim 23.7\%$ of the total from simulations. Here, we point out that the experimental data of sound intensity are the post-processed results of the deconvolution of the measured intensity distribution and the aperture function of a finite-size hydrophone. The

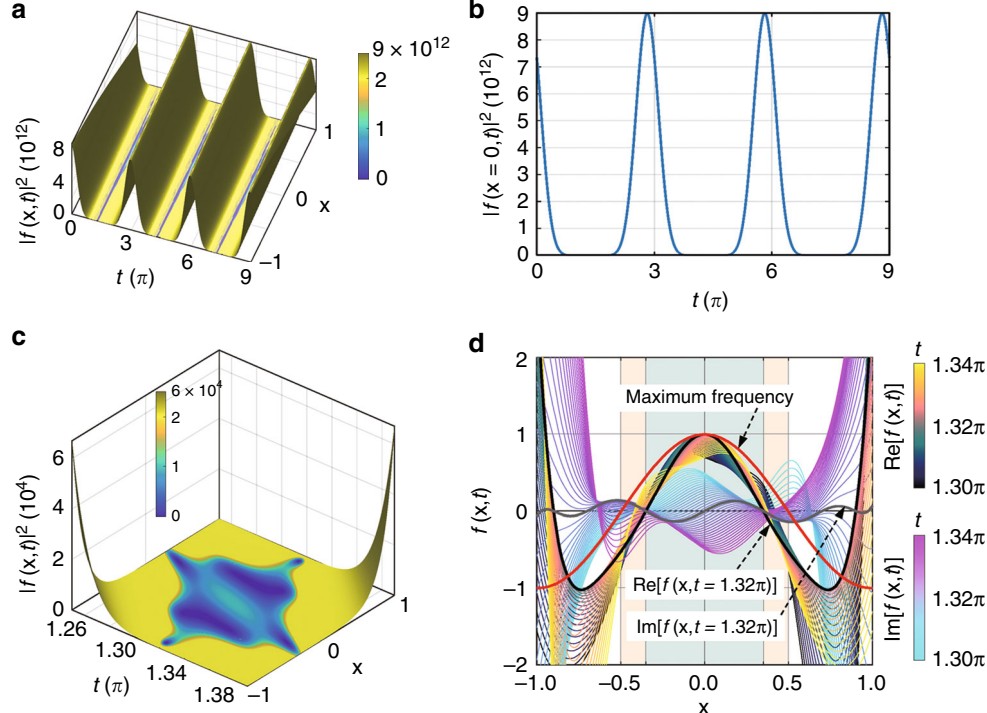

**Fig. 1** Construction of super-oscillatory packets. **a** Revivals of the super-oscillation wave function in time. **b** The distribution of the wave function $|f(x,t)|^2$ along $x = 0$. **c** Zooming-in display of the super-oscillatory region when $t = 1.26\pi \sim 1.38\pi$. **d** Real and imaginary parts of this super-oscillation wave function when $t = 1.3\pi \sim 1.34\pi$. The pseudo-colors in the upper (lower) bars represent the evolving time for the real (imaginary) parts of the wave function

details of deconvolution are described in Supplementary Note 1. In Fig. 3e, we quantitatively compare the simulated and measured field intensity along the line $x = [-\lambda, \lambda]$, $y = 0$, $z = 5.2\lambda$, as displayed by the black solid line and blue circles with the error bars, respectively. We can clearly see that the measured data are in well agreement with the simulation results in Fig. 3. Nevertheless, the recorded focal spot with the radius $\sim 0.3\lambda$ provides us the evidence that a super-oscillatory ultrasound field is created at the far-field, in light of the super-oscillation criterion that $0.3\lambda < \lambda_D = 0.38\lambda$. This super-oscillation effect can also be reflected from the calculated local wavenumber, which is much larger than the one of the maximum spatial frequency component at the zero-intensity position (Supplementary Note 2).

**Force effect of super-oscillation packets**. In nature, there exists an interesting trade-off that one physical quantity varies in an extremely rapid manner by sacrificing the amplitude of another physical quantity. A typical example is the property of singularity point such as vortex[33,34]. At the center of vortex, the phase changes by a rate up to infinity, while the field intensity approaches zero. The same rule also imposes a restriction on super-oscillation, for which the amplitude is small. Thus, the radiation force was intuitively thought to be trivial. Here, we show that the rapidly changed pressure field in super-oscillation packets will generate a distinct pressure gradient and thus a non-trivial time-averaged radiation force, as reflected from the short response time of object trapping. Figure 4a shows the calculated radiation force field where the color bar and the direction of arrows reveal the amplitude and direction of force vectors in the field of view ($2\lambda \times 2\lambda$) on the $x$-$y$ plane at $z = 5.2\lambda$. In Fig. 4a, we set the center point as the coordinate origin and define the radiation force be positive and negative when force vectors are pointing outward and inward, respectively. Figure 4b provides a quantitative comparison between the simulated and measured

radiation force distributions along the line $x = [-\lambda, \lambda]$, $y = 0$, $z = 5.2\lambda$. When $x \geq 0$, we find that there exist three zero-value points of radiation forces. The second one at $x = 0.3\lambda$ is generated via super-oscillation, where the radiation force effect performs like acoustic tweezers[35–38], leading to particle trapping in a steady potential well. However, at the first and third zero-value points locating at $x = 0$ and $x = 0.63\lambda$, the force effect is a push-and-pull instead of pinching. Therefore, particles near the first and third zero-value points are inclined to move toward the second or fourth zero-value points.

We conduct the particle trapping experiment to visually display the profile of the diffraction-limit-broken spot in super-oscillation packets. Figure 4c shows the voltage change applied to the transducer. Figure 4d–g are the photographs taken at different times in Supplementary Video 2, when the voltages are 0, $V_m/2$, $V_m$, and 0, respectively. In Fig. 4d, the particles are randomly distributed in the field of view without radiation force. When the voltage is applied, the radiation force acts on the particles, see Fig. 4e, f. As a result, the particles move towards the second zero-value point at $x = 0.3\lambda$. Figure 4f clearly reveals that tiny particles are tightly squeezed by the super-oscillation tweezer to form a compact ring structure with the radius $\sim 0.3\lambda$. When the power is off, the aggregated granular ring instantly breaks with all the particles randomly distributed in the field of view, as shown in Fig. 4g. Here, we emphasize that the super-oscillation tweezing is very robust and reproducible under a series of voltage pulses with a short response time of <30 ms, as revealed in the recorded video.

**Super-resolution imaging by using super-oscillation packets**. Utilizing the super-oscillation packets, we further demonstrate the super-resolution ultrasound imaging of deep-subwavelength objects in water. In the experiment, the planar sheet with specific patterns is placed in the focal plane ($x$-$y$) of the super-oscillation meta-lens, which is moved in the $x$-$y$ plane by a

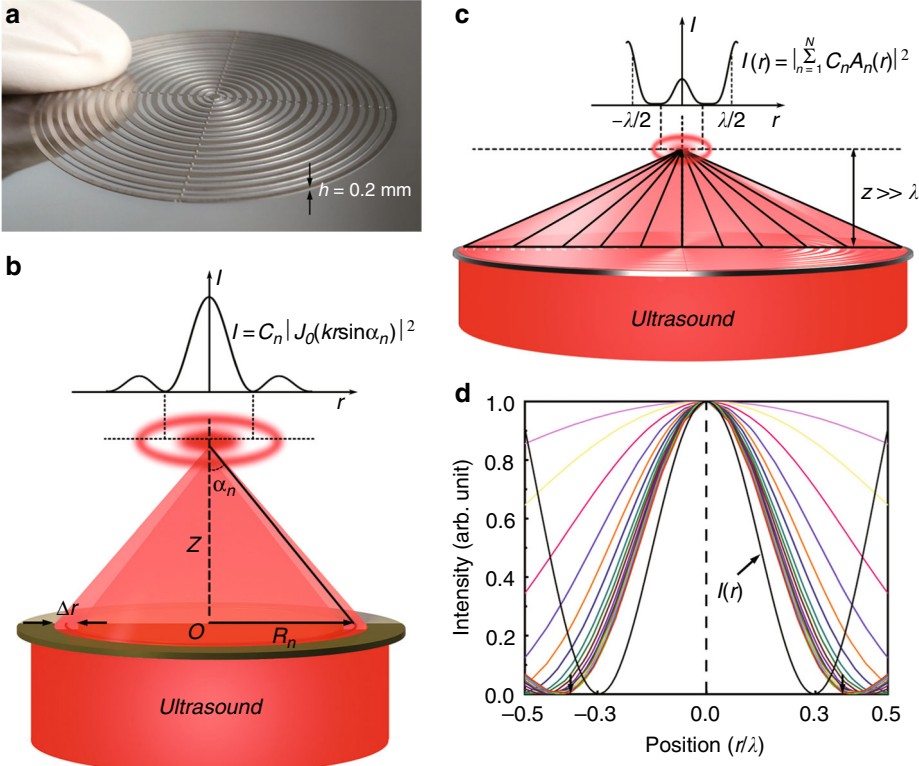

**Fig. 2** Super-oscillation ultrasonic meta-lens. **a** A photograph of the ultrasonic meta-lens. The thickness of this meta-lens is only 0.2 mm, that is, $\sim 0.13\pi$ at the operation frequency of 1 MHz. **b** The ultrasound diffraction from a single-belt Fresnel zone plate with the radius and width marked by $R_n$ and $\Delta r$. The field intensity in the focal plane $I = C_n|J_0(kr\sin\alpha_n)|^2$, where the size of the principal lobe is determined by the diffraction limit ($0.38\lambda$). **c** Ultrasound diffraction from a multiple-belt meta-lens. A radially super-oscillating field pattern is generated beyond the evanescent region ($z\gg\lambda$). **d** Normalized intensity distributions in the focal plane. The pseudo-colored curves correspond to different spatial frequency components. The black curve is the super-oscillation packet that beats the in-plane diffraction limit, which is generated by superimposing those components. The bottom arrows in **d** indicate the location of the diffraction limit ($0.38\lambda$)

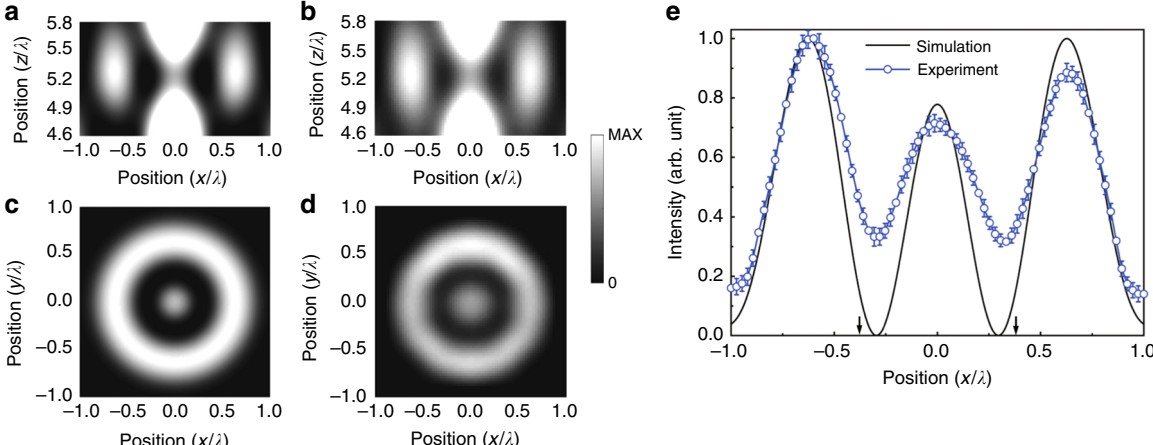

**Fig. 3** Super-oscillation packet in the ultrasonic meta-lens. **a, b** The simulated and measured intensity distributions of a super-oscillatory ultrasound field on the x-z plane. **c, d** The simulated and measured intensity distributions of the super-oscillation packet in the field of view ($2\lambda \times 2\lambda$) on the x-y plane at $z = 5.2\lambda$. **e** The simulated and measured intensity distributions along the line at $x = [-\lambda, \lambda]$, $y = 0$, $z = 5.2\lambda$. The ultrasound frequency is 1 MHz. In **a–e**, the data are normalized with respect to the maximum. The bottom arrows in **e** indicate the location of the diffraction limit ($0.38\lambda$). The experimental error bar is defined by the standard deviation of sound intensity measured in ten times

three-dimensional precision moving stage. In the measurement, the hydrophone and the meta-lens are fixed. Here, it needs to be mentioned that there exists a trade-off that the side-lobe intensity increases exponentially as the main-lobe region oscillates faster[39]. Therefore, we use a thin metal sheet etched with a 0.9 mm-diameter hole to filter the side-lobes in the super-oscillatory field. To demonstrate the super-resolution ultrasound imaging, we fabricate three patterns with subwavelength features on a 0.2 mm-thick steel sheet via the molecular etching technology, as shown in Fig. 5a. We first demonstrate the super-resolution imaging of double slits, where the slit width and the gap between the double slits are 0.4 mm. The gap size is much smaller than the Rayleigh

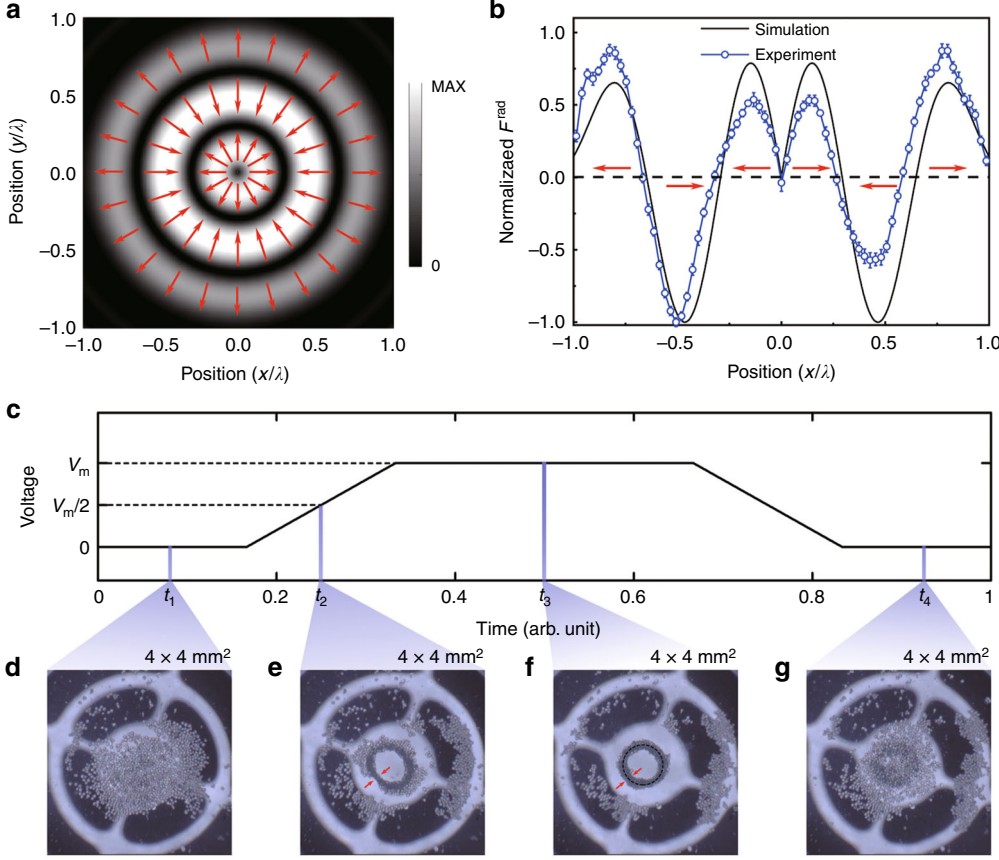

**Fig. 4** Super-oscillation ultrasound tweezing. **a** The simulated radiation force distribution in the super-oscillatory field. The force acts on polystyrene particles with the compressibility $\kappa_p = 2.38 \times 10^{-10}$ Pa$^{-1}$, the longitudinal wave velocity $c_p = 2350$ m s$^{-1}$, the density $\rho_p = 1050$ kg m$^{-3}$ and the mean diameter $a = 100$ μm. **b** The simulated and measured radiation force distributions along the line $x = [-\lambda, \lambda]$, $y = 0$, $z = 5.2\lambda$. In **a** and **b**, directions of force vectors are marked by the red arrows. **c** The schematic voltage change with time. **d–g** The particle distributions at four different times, that is, $t_1$, $t_2$, $t_3$, and $t_4$ as marked in **c**. In **f**, the tightly trapped ring describes the profile of the diffraction-limit-broken spot in super-oscillation packets. The error bar is defined by the standard deviation of normalized radiation force measured in ten times

diffraction limit $0.61\lambda$, with $\lambda = 1.5$mm at 1 MHz. From Fig. 5b, the gap is well resolved by our designed super-oscillation meta-lens. Using a periodic-belt Fresnel zone plate, however, the imaging gap is less distinguishable. In Fig. 5b, a quantitative comparison of the intensity profiles across the center of double slits provides us the evidence of the dramatic improvement of the imaging resolution, where the gap zone corresponds to the central dip of the intensity line (red curve). Then, to demonstrate the performance of the super-oscillation meta-lens in imaging complex objects, we use the pattern of a coiled slit, where the widths of gaps and slits are also 0.4 mm. Comparing the images in Fig. 5c, d, we find that the meta-lens outperforms the periodic-belt Fresnel zone plate in distinguishing subwavelength features, i.e., coiled slits and gaps. Finally, we image a hole array to check how small the hole can be discerned by this technique. The diameter of the smallest holes is 0.4 mm. Figure 5e, f show the imaging results. In Fig. 5f, the holes are blurry and the smallest ones cannot be resolved via the periodic-belt Fresnel zone plate. We can barely tell the bigger holes with the diameter ~1.2 mm. In contrast, the image taken by the super-oscillation meta-lens is impressive with all major features sharp and resolvable, as shown in Fig. 5e. In the end, we point out that our work is in stark contrast with the previous studies[40], which did not demonstrate subwavelength focusing in the critical super-oscillation regime ($<0.38\lambda$)[32,39]. Furthermore, those studies did not present the advanced technique of super-resolution imaging based on the subwavelength focusing[40,41].

## Discussion
In this work, we investigate the super-oscillation packets in an ultrasonic meta-lens, reshaping ultrasound at frequencies that surpass its highest Fourier component. We design a meta-lens that projects different spatial frequency components to generate super-oscillation packets in the far-field. Utilizing the optimization-free meta-lens, we observe super-oscillation ultrasound packets that break the Rayleigh diffraction limit in the far-field. We explore the force effect of oscillation packets, where the profile of a diffraction-limit-broken spot beyond the evanescent region is visually presented by the ring-shaped trapping of tiny particles. We further show the results of super-resolution imaging of various complex patterns that carry subwavelength features. Our work opens a new scenario in the field of ultrasonics. For example, super-oscillation technique brings us flexible controls of ultrasound beyond the diffraction limit in the far-field, which may revolutionize the fields of ultrasonic therapy and imaging.

## Methods
**Design of the super-oscillation meta-lens**. The complex amplitude at an arbitrary point in space can be calculated by the Kirchhoff–Helmholtz integral, which is simplified to the scalar Rayleigh–Sommerfeld diffraction integral[41,42]. The diffraction field of a single-belt Fresnel plate on the target plane $(r, \theta, z)$ is

$$p_n(r, \theta, z) = \frac{1}{2\pi} \int_{R_n - \Delta r/2}^{R_n + \Delta r/2} \int_0^{2\pi} p_n(r', \theta', 0) \frac{z \exp(iks)}{s^2} (ik - \frac{1}{s}) r' dr' d\theta' \quad (5)$$

where $s = \sqrt{(r\cos\theta - r'\cos\theta')^2 + (r\sin\theta - r'\sin\theta')^2 + z^2}$ is the distance

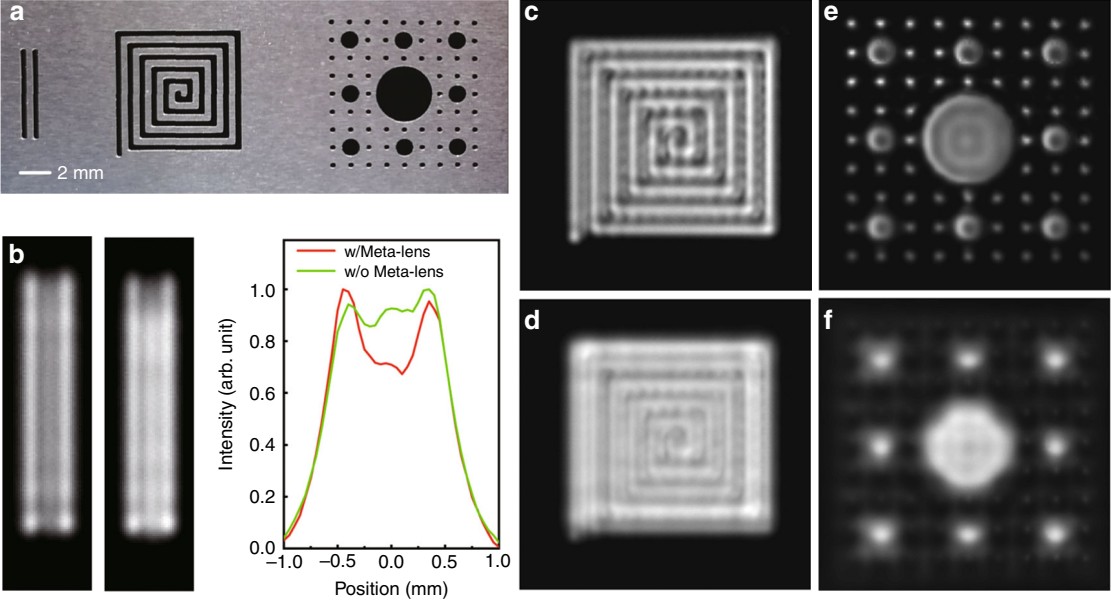

**Fig. 5** Super-resolution ultrasound imaging. **a** The photograph of the mask sheet carved with specific patterns. **b** The ultrasound image of double slits. Left: super-resolution imaging via a meta-lens. Middle: conventional imaging via a periodic-belt Fresnel zone plate. Right: comparison of the intensity profiles across the center of double slits. The slit width and the spacing between double slits are both 0.4 mm. **c** Super-resolution image of a coiled slit by using the meta-lens. **d** Conventional image of the coiled slit by using the periodic-belt Fresnel zone plate. **e** Super-resolution image of a hole array by using the meta-lens. **f** Conventional image of the hole array by using the periodic-belt Fresnel zone plate

between the source point and the target point, and $k = \omega/c$ the wavenumber. Solving Eq. (5), the intensity in the focal region is $I = C_n |J_0(kr \sin \alpha_n)|^2$, where $\sin \alpha_n = R_n / \sqrt{R_n^2 + z^2}$, see Fig. 2b. For a multiple-belt meta-lens, the diffraction field can be written as $p(r) = \sum_{n=1}^{N} C_n A_n(r)$ by setting the amplitude on the target plane $A_n(r) = p_n(r)/C_n$ with $C_n = p_n(0)$. The spatial frequency $\eta_n = \sin \alpha_n/\lambda$, see Fig. 2c. Discretizing the super-oscillatory function that satisfies $p(l_m) = p_m$ at the positions $r = \{l_m\}_{m=1}^{M}$ on the target plane, we obtained the matrix equation

$$SC = P \qquad (6)$$

where $S$ is a $M \times N$ matrix with the element $S_{mn} = A_n(l_m)$, the weighted coefficient $C = (C_1, C_2, \cdots, C_N)^T$ and $P = (p_1, p_2, \cdots, p_M)^T$. Equation (6) has a solution at $M \leq N$. Here, we have $M = N$. Therefore, Eq. (6) has a unique solution. Since the elements $S_{mn}$ and $C_n$ are only dependent on the unknown parameters $R_n$, with the other parameters $\Delta r$ and $z$ all fixed constants, it is thus a nonlinear problem to solve Eq. (6) for $R_n$. In our preset design, the principal lobe of super-oscillation packets has the radius of $0.3\lambda$ beyond the diffraction limit ($<0.38\lambda$), and locates on the plane $z = 5.2\lambda$. The widths of slits are fixed at $\Delta r = \lambda/2$. Based on the optimization-free approach of Eq. (6), we directly obtain the values of $R_n$, where the frequency $f = 1$MHz, $R_n = (rr_n + r_n)/2$ and $\Delta r = rr_n - r_n$. Here, the structure parameters of the meta-lens $\{r_n\}_{n=1}^{N}$ and $\{rr_n\}_{n=1}^{N}$ are the inner and outer radii of the circular slits, Supplementary Fig. 1. Based on these geometrical parameters, we fabricated the meta-lens via the technique of molecular etching.

**Numerical simulation**. We employ the acoustic-solid interaction module in COMSOL Multiphysics$^{TM}$ 5.3a to perform full-wave simulations. The geometrical parameters of models were appended in Supplementary Table 1. For the metal steel, the density $\rho = 7760$kg m$^{-3}$, the longitudinal wave velocity $c_l = 6010$m s$^{-1}$ and the shear wave velocity $c_t = 3320$m s$^{-1}$. For water, the density $\rho_0 = 1000$kg m$^{-3}$ and the sound velocity $c_0 = 1500$m s$^{-1}$. Perfectly matched layers are imposed at the outer boundaries of simulation domains to prevent reflection.

**Experiment set-ups and measurements**. In experiment, we conduct measurements in a water tank ($500 \times 600 \times 1000$ mm$^3$), Supplementary Fig. 2. In order to eliminate the influence of reflected waves, we output an electrical pulse series of sinusoidal signals via a multi-function waveform generator (Tektronix AFG3022C). We use a connected transducer (Harisonic I3-0108-P) to convert the electrical signal into ultrasound packets. The ultrasound field is scanned by a 0.5 mm-diameter needle hydrophone. All the data are recorded into a digital storage oscilloscope (Agilent Technologies DSO-X-3034A) and processed with the Precision Acoustics UMS3 system software. Since water in the tank is purified by the purification system, the effects of losses and scatterings on the super-oscillation phenomena are neglected in this work. In the presence of randomly distributed

scatterers before the target plane, a focused super-oscillation field could still be obtained by adding a phase-amplitude modulator based on the time reversal technique[43]. In radiation force experiment, we use a smaller water container ($400 \times 300 \times 450$ mm$^3$) and control a transfer pipette to distribute sphere particles of mean diameter ~100 μm in the auxiliary frame $5.2\lambda$ away from the meta-lens, Supplementary Fig. 3a. A video is recorded to monitor the real-time motion of those micro-particles via a stereomicroscope (ZEISS V20). The peak value of voltage applied to the transducer is $V_m = 0.12$ V with the use of a stable high-power amplifier E&I 2200 L. The imaging experiment is shown in Supplementary Fig. 3b.

**Calculation of acoustic radiation force**. The radiation force exerted on a polystyrene particle is calculated by solving the vector surface integral of the time-averaged wave momentum flux over a surface $\Omega$ enclosing the particle. The polystyrene particle has the compressibility $\kappa_p = 2.38 \times 10^{-10}$ Pa$^{-1}$, the longitudinal wave velocity $c_p = 2350$ m s$^{-1}$, the density $\rho_p = 1050$ kg m$^{-3}$ and the mean diameter $a = 100$ μm. We define $\mathbf{n}$ as the unitary normal vector pointing away from $\Omega$. Within the second-order approximation, we have the expression of acoustic radiation force[44–49]

$$\mathbf{F}^{rad} = -\int_{\Omega} \left[ \langle \sigma_2 \rangle + \rho_0 \langle \mathbf{v}_1 \mathbf{v}_1 \rangle \right] \cdot \mathbf{n} ds \qquad (7)$$

where $\mathbf{v}_1$, $\sigma_2$ are the first-order velocity (proportional to the pressure gradient) and the second-order stress at the particle surface. The bracket symbol denotes the time average over one oscillation period. The deduction details of Eq. (7) and its simplified form are shown in Supplementary Notes 3 and 4. Finally, substituting the measured sound pressure on the imaging plane into the simplified form of Eq. (7), we will obtain the acoustic radiation force of polystyrene particle as shown in Fig. 4b.

## Data availability
The datasets generated during the analyses during this study are available from the corresponding authors on reasonable request.

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

## Acknowledgements

This work was supported by National Natural Science Foundation of China (Grant Nos. 11674119, 11674346, 11534013, 11690030, 11690032). F. C. was partially supported by the Shenzhen Basic Research Program (Grant No. JCYJ20170818163258397). K.H. thanks CAS Poineer Hundred Talents Program, "The Fundamental Research Funds for the Central Universities" in China, the National Nature Science Foundation of China (Grant No. 61875181 and 61705085), and the support from the University of Science and Technology of China's Center for Micro and Nanoscale Research and Fabrication. X.F.Z., Y.X.S., and Y.G.P. was supported by the Fundamental Research Funds for the Central Universities (Grant No. 2019kfyRCPY136).

## Author contributions

X.F.Z., C.W.Q., F.C., and H.Z. conceived and designed the experiments. Y.X.S. and Y.G.P. fabricated the samples. Y.X.S. and F.C. performed the experiments. Y.X.S., K.H. and Y.G.P. performed simulations and analyzed the data. All authors wrote the paper.

## Additional information

**Competing interests:** The authors declare no competing interests.

