## [Peer Review File · Nature Communications]

Reviewers' Comments:

Reviewer #1:

Remarks to the Author:

The authors reported the generation of ultrasonic super-oscillation wave-packets using an 'acoustic metasurface'. The optimization-free method to construct the super-oscillation wave-packets is clearly described, and its validity is confirmed by the reasonable agreement between the simulated and experimentally recorded field profiles. The calculation of acoustic radiation force based on solving the governing equation in liquid is new and inspiring, and therefore vital for generalized study on similar problems.

Moreover, the ultrasonic trapping and imaging experiments are very impressive, where the visualization of enhanced pressure gradient (or acoustic radiation force) and imaging resolution improvement show great advantages over previous methods.

Overall, this paper is well written and organized. The design, data analysis, experimental characterization and discussion are clear. I intend to recommend this work to be accepted for publication on Nature Communications if the authors can address the following issues:

1. The terminology 'metasurface' generally refers to artificial thin structures involving specific electromagnetic resonance to tune the amplitude, phase or both. The thickness of 'meta-lens' in this work is indeed deep subwavelength (0.13λ), but it seems to be a binary amplitude mask without any EM resonance. So I am not convinced to use 'metasurface' throughout the text. Please comment.
2. It is well known that a super-oscillation function happens near the perfect destructive interference with low intensity. But the main experimental super-oscillation wave-packet presented in Fig. 3 gives a nonzero intensity near the first minimum compared with the close-zero intensity distribution in the simulation (even much higher than the second intensity minimum by observing the trend of intensity decrement, blue curve in Fig. 3e). The authors explained that this deviation is 'partly due to the limited sensitivity and finite size of the hydrophone'. To convince it further, can the authors simulate the convolution of the ideal super-oscillatory function (black curve in Fig. 3e) and the aperture function (0.5 mm-diameter) of the experimentally used hydrophone, and then compare with the experimental results? If they are similar, it will be reasonable to use a de-convolution process to calculate the field distributions in Fig. 3 and radiation force distributions in Fig. 4b.
3. How is the experimental error bar in Fig. 3e defined? Please make it clear.
4. The local wavenumber near the super-oscillation region in Fig. 3 should be given to make sure it is larger than the maximum spatial frequency of the constituent wave.
5. Apart from the three main peaks shown in Fig. 3, are there any other significant sidelobes out of the field of view [-1.5mm, 1.5mm]? This should be briefly discussed for estimation of the overall focusing performance of the device.
6. The main concern is the imaging control experiment in Fig. 5. The imaging capability with the acoustic 'meta-lens' is convincing by looking at the results in Figs. 5c & 5f. The authors clearly stated that a 'single-belt Fresnel zone plate' is used for the control experiment, while its parameters are not given. What's the dimension, working distance and point-spread-function of the single-belt Fresnel zone plate used? For a fair comparison with a super-oscillation acoustic 'meta-lens' and to show the resolution improvement factor, a focusing element (Fresnel zone plate, or others) with similar aperture size and working distance (determining the effective numerical aperture) should be generally used. It's much more reasonable to use a multi-belt Fresnel zone plate rather than a single-belt one here.

Reviewer #2:

Remarks to the Author:

In this manuscript, the authors realize sub-wavelength focusing of ultrasonic waves in the far-field by utilizing super-oscillation wave-packets emitted by the acoustic metasurfaces. The ring-shaped trapping of tiny particles and super-resolution ultrasound imaging have also been experimentally

verified. These results are interesting, meaningful, and might be considered to be accepted by Nature Communications, if the authors can consider the following problems.

(1) In some previous investigations, super-oscillation wave-packets and sub-wavelength focusing for the ultrasonic wave have been theoretically studied (such as in Sci. Rep. 4, 6257 (2014)) and experimentally realized (such as in Sci. Rep. 8, 9131 (2018)). The present work is closely related to the previous investigations. The authors should point out the differences and advantages in this work, compared to these previous investigations.

(2) In Figure. 3e, the intensity of the side-lobe is higher than the main-lobe. How to avoid the influences of the side-lobes? Can the side-lobe be further suppressed by optimizing the structural parameters of the metasurfaces?

(3) In Fig. 4b, the simulation and experiment results of the radiation force distributions are presented. The authors should give more details about the measurement of the radiation force.

(4) There are always some losses or unwanted scatterings when the waves propagate to the target plane. The authors should emphasize the effects of losses and scatterings on the phenomena.

Response to Referee 1's comments

The authors reported the generation of ultrasonic super-oscillation wave-packets using an 'acoustic metasurface'. The optimization-free method to construct the super-oscillation wave-packets is clearly described, and its validity is confirmed by the reasonable agreement between the simulated and experimentally recorded field profiles. The calculation of acoustic radiation force based on solving the governing equation in liquid is new and inspiring, and therefore vital for generalized study on similar problems. Moreover, the ultrasonic trapping and imaging experiments are very impressive, where the visualization of enhanced pressure gradient (or acoustic radiation force) and imaging resolution improvement show great advantages over previous methods. Overall, this paper is well written and organized. The design, data analysis, experimental characterization and discussion are clear. I intend to recommend this work to be accepted for publication on Nature Communications if the authors can address the following issues:

Response: We sincerely thank the reviewer for the positive remarks and valuable suggestions that helps us to improve the manuscript.

1. The terminology 'metasurface' generally refers to artificial thin structures involving specific electromagnetic resonance to tune the amplitude, phase or both. The thickness of 'meta-lens' in this work is indeed deep subwavelength (0.13λ), but it seems to be a binary amplitude mask without any EM resonance. So I am not convinced to use 'metasurface' throughout the text. Please comment.

Response: We have replaced the ‘meta-surface’ with ‘meta-lens’ throughout the text.

2. It is well known that a super-oscillation function happens near the perfect destructive interference with low intensity. But the main experimental super-oscillation wave-packet presented in Fig. 3 gives a nonzero intensity near the first minimum compared with the close-zero intensity distribution in the simulation (even much higher than the second intensity minimum by observing the trend of intensity decrement, blue curve in Fig. 3e). The authors explained that this deviation is ‘partly due to the limited sensitivity and finite size of the hydrophone’. To convince it further, can the authors simulate the convolution of the ideal super-oscillatory function (black curve in Fig. 3e) and the aperture function (0.5 mm-diameter) of the experimentally used hydrophone, and then compare with the experimental results? If they are similar, it will be reasonable to use a de-convolution process to calculate the field distributions in Fig. 3 and radiation force distributions in Fig. 4b.

Response: Thank you for your very good suggestion. Yes, the deconvolution operation further improves our experimental results, as shown by the revised Fig. 3 and Fig. 4b.

Please refer to the lines 124-128 in the revised main text. *“Here we point out that the experimental data of sound intensity are the post-processed results of the deconvolution of the measured intensity distribution and the aperture function of a finite-size hydrophone. The details of deconvolution are described in Supplementary Note 1.”*

Figure 3 | Super-oscillation packet in the ultrasonic meta-lens. *a, b*, The simulated and measured intensity distributions of a super-oscillatory ultrasound field on the x - z plane. *c, d*, The simulated and measured intensity distributions of the super-oscillation packet in the field of view ($2\lambda \times 2\lambda$) on the x - y plane at $z=5.2\lambda$. *e*, The simulated and measured intensity distributions along the line at $x=[-\lambda, \lambda]$, $y=0$, $z=5.2\lambda$. The ultrasound frequency is 1 MHz. In (a)-(e), the data are normalized with respect to the maximum. The bottom arrows in *e* indicate the location of the diffraction limit (0.38λ).

Figure 4 | Super-oscillation ultrasound tweezing. *a*, The simulated radiation force distribution in the super-oscillatory field. The force acts on polystyrene particles with

the compressibility $\kappa_p = 2.38 \times 10^{-10} \text{ Pa}^{-1}$, the longitudinal wave velocity $c_p = 2350 \text{ m}\cdot\text{s}^{-1}$, the density $\rho_p = 1050 \text{ kg}\cdot\text{m}^{-3}$ and the mean diameter $a = 100 \text{ }\mu\text{m}$. **b**, The simulated and measured radiation force distributions along the line $x = [-\lambda, \lambda]$, $y = 0$, $z = 5.2\lambda$. In **a** and **b**, directions of force vectors are marked by the red arrows. **c**, The schematic voltage change with time. **d-g**, The particle distributions at four different times, that is, t_1 , t_2 , t_3 , and t_4 as marked in **c**. In **f**, the tightly trapped ring describes the profile of the diffraction-limit-broken spot in super-oscillation packets.

Also in the supplementary part, please refer to

“Supplementary Note 1 | Convolution and deconvolution

In mathematics, the convolution between the functions $f(x, y)$ and $h(x, y)$ is defined by

$$\begin{aligned} g(x, y) &= f(x, y) * h(x, y) \\ &= \iint f(\xi, \eta) h(x - \xi, y - \eta) d\xi d\eta, \end{aligned} \quad (S1)$$

where $*$ denotes the convolution operator, $f(x, y)$ is the source function before the convolution process, $h(x, y)$ is the convolutional interaction function and $g(x, y)$ is the solution function after convolution. In experimental measurements, the functions $f(x, y)$, $h(x, y)$ and $g(x, y)$ are discretized into matrices. Then the Eq. (S1) can be expressed into

$$g(s, t) = \sum_{m=0}^{F_r-1} \sum_{n=0}^{F_c-1} f(m, n) h(s - m, t - n), \quad (S2)$$

where (m, n) and (s, t) denote the element indices of the matrices $f_{F_r \times F_c}$ and $h_{H_r \times H_c}$. F_r and F_c represent the numbers of rows and columns of the source function

matrix $f_{F_r \times F_c}$. The pressure field scanning is actually a convolution process, where the measured field distribution can be regarded as the solution function $g(x, y)$. The aperture function of hydrophone is the convolutional interaction function $h(x, y)$, which is a truncated Gaussian function with the FWHM ~ 2 mm (size of the truncated region: ~ 0.6 mm). The source function $f(x, y)$ is the one that reflects the real pressure field distribution, which can be solved by the deconvolution process, as schematically described in Figs. S5 and S6.”

Figure S5 | The deconvolution in pressure field post-processing. In experimental measurements, the measured field distribution is the solution function $g_{G_r \times G_c}$. The aperture function of hydrophone is the convolutional interaction function $h_{H_r \times H_c}$. The real pressure field distribution is the source function $f_{F_r \times F_c}$.

Figure S6 | The deconvolution in deciphering the super-oscillation field. Left: the post-processed intensity field of the super-oscillation packet. Middle: the aperture function of hydrophone. Right: the measured intensity field in experiments, which is actually the convolution between the post-processed field and the aperture function.

3. How is the experimental error bar in Fig. 3e defined? Please make it clear.

Response: Thank you for raising this important point. In experiments, we measured the intensity field ten times. The experimental data are the average of all the measurements, where the experimental error bar in Fig. 3e is defined by the standard deviation of the data sets.

Please refer to the lines 128-131 in the revised main text. “In Fig. 3e, we quantitatively compare the simulated and measured field intensity along the line $x=[-\lambda, \lambda], y=0, z=5.2\lambda$, as displayed by the black solid line and blue circles with the error bars, respectively. The experimental error bar is defined by the standard deviation of sound intensity measured in ten times.”

4. The local wavenumber near the super-oscillation region in Fig. 3 should be given to make sure it is larger than the maximum spatial frequency of the constituent wave.

Response: Thank you for this very good suggestion. In the revised version, we have provided the local wavenumber in Fig. S7b.

Please refer to the lines 135-138 in the revised main text. “*This super-oscillation effect can also be reflected from the calculated local wave number, which is much larger than the one of the maximum spatial frequency component at the zero-intensity position (Supplementary Note 2).*”

Please refer to the lines 90-103 in the revised supplementary information. “

Supplementary Note 2 | Local wavenumber near the super-oscillation region

For a single-belt Fresnel zone plate, the intensity distribution at the focal region is described by $I = C_n |J_0(kr \sin \alpha_n)|^2$, where the first minimum locates at $r = 0.38\lambda / \sin \alpha_n$. Since $\sin \alpha_n \leq 1$, the diffraction limit is thus $r_D = 0.38\lambda$. Figure S7a shows the normalized intensity distributions of the super-oscillation field (black line) and the maximum spatial frequency component (red line). The recorded focal spot with the radius $\sim 0.3\lambda$ provides us a clear evidence that a super-oscillatory field is created, based on the super-oscillation criterion that $0.3\lambda < \lambda_D = 0.38\lambda$. Moreover, from the definition of local wavenumber $k(r) = \text{Im}\{\partial_r[\ln p(r)]\}$, where $p(r)$ is the band-limited function, we calculated the local wavenumbers of the super-oscillation band-limited function (black line) and the maximum spatial frequency component (red line) near the super-oscillation region ($r = [-0.35\lambda, 0.35\lambda]$), as shown in Fig. S7b. The

results show that the local wavenumber of super-oscillation is much larger than the one of maximum spatial frequency component at the zero-intensity position.”

Figure S7 | The intensity curves and local wavenumbers. *a*, The normalized intensity distributions of a super-oscillatory field (black line) and the maximum spatial frequency component (red line). The bottom arrows indicate the location of the diffraction limit (0.38λ). *b*, The calculated local wavenumbers of a super-oscillatory field (black line) and the maximum spatial frequency component (red line).

5. Apart from the three main peaks shown in Fig. 3, are there any other significant side-lobes out of the field of view $[-1.5\text{mm}, 1.5\text{mm}]$? This should be briefly discussed for estimation of the overall focusing performance of the device.

Response: Thank you for this comment. Yes, there exist other side-lobes outside the field of view ($2\lambda \times 2\lambda$), as shown in Fig. R1. We add some discussions in the main text about the focusing performance of the device.

Please refer to the lines 122-124 in the revised main text. “Figures 3c and 3d present the simulated and measured intensity profiles in the field of view ($2\lambda \times 2\lambda$) on the x - y plane at $z=5.2\lambda$, respectively. Outside the field of view on the x - y plane, the acoustic

energy accounts for ~23.7% of the total from simulations.”

Figure R1. The intensity distribution of a super-oscillation field on the x - y plane.

6. The main concern is the imaging control experiment in Fig. 5. The imaging capability with the acoustic ‘meta-lens’ is convincing by looking at the results in Figs. 5c & 5f. The authors clearly stated that a ‘single-belt Fresnel zone plate’ is used for the control experiment, while its parameters are not given. What’s the dimension, working distance and point-spread-function of the single-belt Fresnel zone plate used? For a fair comparison with a super-oscillation acoustic ‘meta-lens’ and to show the resolution improvement factor, a focusing element (Fresnel zone plate, or others) with similar aperture size and working distance (determining the effective numerical aperture) should be generally used. It’s much more reasonable to use a multi-belt Fresnel zone plate rather than a single-belt one here.

Response: Thank you for this very important suggestion. For the single-belt Fresnel zone plate, the radius $R_n = 7.5$ mm , the width $\Delta r = 0.75$ mm , the imaging plane

locates at $z = 5.2\lambda$. In light of your suggestion, we fabricated a periodic-belt Fresnel zone plate with the similar aperture size and working distance for a fair comparison, as shown in Fig. 5 and Fig. S4.

Figure 5 | Super-resolution ultrasound imaging. *a*, The photograph of the mask sheet carved with specific patterns. *b*, The ultrasound image of double slits. Left: super-resolution imaging via a meta-lens. Middle: conventional imaging via a periodic-belt Fresnel zone plate. Right: comparison of the intensity profiles across the center of double slits. The slit width and the spacing between double slits are both 0.4 mm. *c*, Super-resolution image of a coiled slit by using the meta-lens. *d*, Conventional image of the coiled slit by using the periodic-belt Fresnel zone plate. *e*, Super-resolution image of a hole array by using the meta-lens. *f*, Conventional image of the hole array by using the periodic-belt Fresnel zone plate.

Figure S4 | The imaging comparison of different lenses. *a*, The fabricated samples of a single-belt lens (left), a periodic-belt lens (middle) and a super-oscillation meta-lens (right). *b*, Ultrasound images of a coiled slit. Left: conventional imaging via a single-belt Fresnel zone plate. Middle: conventional imaging via a periodic-belt Fresnel zone plate. Right: super-resolution imaging via a meta-lens. *c*, Ultrasound images of a hole array. Left: conventional imaging via a single-belt Fresnel zone plate. Middle: conventional imaging via a periodic-belt Fresnel zone plate. Right: super-resolution imaging via a meta-lens. *d*, The intensity distributions along the dashed lines in *b*. *e*, The intensity distributions along the dashed lines in *c*. For the single-belt Fresnel zone plate, the radius $R_n = 7.5$ mm the width $\Delta r = 0.75$ mm the imaging plane locates at $z = 5.2\lambda$ with the focused sound intensity satisfying $I = C_n |J_0(kr \sin \alpha_n)|^2$. For the periodic-belt Fresnel zone plate, the period $p = 1.65$ mm the width $\Delta r = 0.75$ mm the imaging plane locates at $z = 5.2\lambda$.

Response to Referee 2's comments

In this manuscript, the authors realize sub-wavelength focusing of ultrasonic waves in the far-field by utilizing super-oscillation wave-packets emitted by the acoustic metasurfaces. The ring-shaped trapping of tiny particles and super-resolution ultrasound imaging have also been experimentally verified. These results are interesting, meaningful, and might be considered to be accepted by Nature Communications, if the authors can consider the following problems.

Response: We sincerely thank the reviewer for the positive remarks and valuable suggestions that helps us to improve the manuscript.

(1) In some previous investigations, super-oscillation wave-packets and sub-wavelength focusing for the ultrasonic wave have been theoretically studied (such as in Sci. Rep. 4, 6257 (2014)) and experimentally realized (such as in Sci. Rep. 8, 9131 (2018)). The present work is closely related to the previous investigations. The authors should point out the differences and advantages in this work, compared to these previous investigations.

Response: We would like to thank the reviewer for bringing this important issue to our attention. The previous works mentioned by the referee have been cited in the Refs. 40 (Sci. Rep. 8, 9131 (2018)), 41 (Sci. Rep. 4, 6257 (2014)) of the revised manuscript. In the revised main text, we point out the differences and advantages of our work as suggested by the reviewer.

Detailed discussions please refer to the lines 203-207 of revised main text. “ *In the*

end, we point out that our work is in stark contrast with the previous studies⁴⁰ which did not demonstrate subwavelength focusing in the critical super-oscillation regime ($< 0.38\lambda$)^{32,39}. Furthermore, those studies did not present the advanced technique of super-resolution imaging based on the subwavelength focusing^{40,41}.”

Besides the super-resolution imaging experiment, we also demonstrate the acoustic tweezing effect of tiny particles in this work by using the focused ultrasonic super-oscillation packets.

(2) In Figure. 3e, the intensity of the side-lobe is higher than the main-lobe. How to avoid the influences of the side-lobes? Can the side-lobe be further suppressed by optimizing the structural parameters of the metasurfaces?

Response: Thank you for pointing out this important issue. For a planar diffraction lens, the lateral size and axial position of its focal spot, in principle, can be customized by optimizing the lens structural parameters (width and radial position of each belt). The traditional Fresnel zone plate can focus sound waves into an Airy spot described by $|J_0(kr\sin\alpha) + J_2(kr\sin\alpha)|^2$, where J_0 and J_2 are the zero-order and second-order Bessel functions of the first kind, $k = 2\pi/\lambda$ the wavenumber, r the radial position, $\sin\alpha$ the numerical aperture (NA). The Airy spot has the main-lobe size of $0.61\lambda/\sin\alpha$ and a weak side-lobe with mere 1.75% intensity of the main-lobe. For a single-belt Fresnel zone plate, as shown in Fig. 2b of the main text, the intensity distribution at the focal region is $|J_0(kr\sin\alpha_n)|^2$, which has the main-lobe size of $0.38\lambda/\sin\alpha_n$, accompanied with moderate side-lobes [Ref. 39]. The peak-intensity

ratio between the strongest side-lobe and the central main-lobe is $\sim 16.2\%$. Since $\sin \alpha_n \leq 1$, the diffraction limit is thus defined by $r_D = 0.38\lambda$.

In this work, we show that the focal spot can be further reduced by adjusting different spatial frequency components in the optimization, so that the pressure field in the main-lobe region oscillates faster than the maximum-frequency component. Therefore, it is reasonable to take $r_D = 0.38\lambda$ as the super-oscillation criterion. However, there exists a trade-off that the side-lobe intensity increases exponentially as the main-lobe region oscillates faster, shown in Fig. R2. Therefore, we need to filter out the side-lobe effect in the super-resolution ultrasound imaging [Ref. 39]. Here, we use a thin metal sheet etched with a 0.9 mm-diameter hole to filter the side-lobes in the super-oscillatory field (see the revised lines 179-182 in the main text).

Figure R2. Possible intensity patterns of focal spots by the planar lens. By Rayleigh criterion (RC) and super-oscillation criterion (SOC), the focal spots are categorized into three gradually variant ranges. From the range of above-resolution to super-oscillation, the size of main-lobe decreases smoothly, accompanied by the gradually increasing side-lobes, as the inset field patterns shown [Ref. 39].

(3) In Fig. 4b, the simulation and experiment results of the radiation force distributions are presented. The authors should give more details about the measurement of the radiation force.

Response: Thank you for this important suggestion. The radiation force is indirectly measured by substituting the measured sound pressure on the imaging plane into the simplified form of the acoustic radiation force equation. In the revised main text and supplementary materials, we give the details.

Please refer to “*The deduction details of Eq. (7) and its simplified form are shown in Supplementary Notes 3 and 4. Finally, substituting the measured sound pressure on the imaging plane into the simplified form of Eq. (7), we will obtain the acoustic radiation force of polystyrene particle as shown in Fig. 4b.*” **on lines 303-306 of the main text.**

The simplified form of Eq. (7) is given in the supplementary information with added information.

Please refer to “*From Eq. (S25), the acoustic radiation force \mathbf{F}^{rad} is expressed in terms of the velocity potential of incident acoustic wave ϕ_{in} at the particle position as well as the scattering coefficients of the monopole and dipole components, namely, f_1 and f_2 . Based on the previous study, the monopole scattering coefficient*

$$f_1 = 1 - \frac{\rho_0 c_0^2}{\rho_p c_p^2} \text{ and dipole scattering coefficient } f_2 = \frac{2(\rho_p - \rho_0)}{2\rho_p + \rho_0}, \text{ where } c_0 \text{ and } \rho_0$$

are the sound speed and the density of water, c_p and ρ_p denote the sound speed and the density of tiny particles.” **on lines 225-231 of the supplementary information.**

(4) There are always some losses or unwanted scatterings when the waves propagate to

the target plane. The authors should emphasize the effects of losses and scatterings on the phenomena.

Response: Following your suggestion, we add some discussions about the effects of losses and scatterings on the super-oscillation phenomena in the revised main text.

Please refer to the lines 280-285 of the revised main text. *“Since water in the tank is purified by the purification system, the effects of losses and scatterings on the super-oscillation phenomena are neglected in this work. In the presence of randomly distributed scatterers before the target plane, a focused super-oscillation field could still be obtained by adding a phase-amplitude modulator based on the time reversal technique⁴³.”*

List of revisions

I. Revisions in the main text

Revisions made to address the reviewers' concerns:

1. In lines 122-138, page 7, revised text

Figures 3c and 3d present the simulated and measured intensity profiles in the field of view ($2\lambda \times 2\lambda$) on the x-y plane at $z=5.2\lambda$, respectively. Outside the field of view on the x-y plane, the acoustic energy accounts for $\sim 23.7\%$ of the total from simulations. Here we point out that the experimental data of sound intensity are the post-processed results of the deconvolution of the measured intensity distribution and the aperture function of a finite-size hydrophone. The details of deconvolution are described in Supplementary Note 1. In Fig. 3e, we quantitatively compare the simulated and measured field intensity along the line $x=[-\lambda, \lambda]$, $y=0$, $z=5.2\lambda$, as displayed by the black solid line and blue circles with the error bars, respectively. The experimental error bar is defined by the standard deviation of sound intensity measured in ten times. We can clearly see that the measured data are in well agreement with the simulation results in Fig. 3. Nevertheless, the recorded focal spot with the radius $\sim 0.3\lambda$ provides us the evidence that a super-oscillatory ultrasound field is created at the far field, in light of the super-oscillation criterion that $0.3\lambda < \lambda_D = 0.38\lambda$. This super-oscillation effect can also be reflected from the calculated local wave number, which is much larger than the one of the maximum spatial frequency component at the zero-intensity position (Supplementary Note 2).

2. In lines 203-207, page 10, added text

In the end, we point out that our work is in stark contrast with the previous studies⁴⁰ which did not demonstrate subwavelength focusing in the critical super-oscillation regime ($<0.38\lambda$)^{32,39}. Furthermore, those studies did not present the advanced technique of super-resolution imaging based on the subwavelength focusing^{40,41}.

3. In lines 280-285, page 14, added text

Since water in the tank is purified by the purification system, the effects of losses and scatterings on the super-oscillation phenomena are neglected in this work. In the presence of randomly distributed scatterers before the target plane, a focused super-oscillation field could still be obtained by adding a phase-amplitude modulator based on the time reversal technique⁴³.

4. In lines 302-305, page 15, revised and added text

The deduction details of Eq. (7) and its simplified form are shown in Supplementary Notes 3 and 4. Finally, substituting the measured sound pressure on the imaging plane into the simplified form of Eq. (7), we will obtain the acoustic radiation force of polystyrene particle as shown in Fig. 4b.

5. In lines 451-478, pages 24-26, revised figures and captions

Figure 3 / Super-oscillation packet in the ultrasonic meta-lens. *a, b*, The simulated and measured intensity distributions of a super-oscillatory ultrasound field on the x - z plane. *c, d*, The simulated and measured intensity distributions of the super-oscillation packet in the field of view ($2\lambda \times 2\lambda$) on the x - y plane at $z=5.2\lambda$. *e*, The simulated and measured intensity distributions along the line at $x=[-\lambda, \lambda]$, $y=0$, $z=5.2\lambda$. The ultrasound frequency is 1 MHz. In (a)-(e), the data are normalized with respect to the maximum. The bottom arrows in *e* indicate the location of the diffraction limit (0.38λ).

Figure 4 | Super-oscillation ultrasound tweezing. *a*, The simulated radiation force distribution in the super-oscillatory field. The force acts on polystyrene particles with the compressibility $\kappa_p = 2.38 \times 10^{-10} \text{ Pa}^{-1}$, the longitudinal wave velocity $c_p = 2350 \text{ m}\cdot\text{s}^{-1}$, the density $\rho_p = 1050 \text{ kg}\cdot\text{m}^{-3}$ and the mean diameter $a = 100 \text{ }\mu\text{m}$. *b*, The simulated and measured radiation force distributions along the line $x = [-\lambda, \lambda]$, $y = 0$, $z = 5.2\lambda$. In *a* and *b*, directions of force vectors are marked by the red arrows. *c*, The schematic voltage change with time. *d-g*, The particle distributions at four different times, that is, t_1 , t_2 , t_3 , and t_4 as marked in *c*. In *f*, the tightly trapped ring describes the profile of the diffraction-limit-broken spot in super-oscillation packets.

Figure 5 | Super-resolution ultrasound imaging. *a*, The photograph of the mask sheet carved with specific patterns. *b*, The ultrasound image of double slits. Left: super-resolution imaging via a meta-lens. Middle: conventional imaging via a periodic-belt Fresnel zone plate. Right: comparison of the intensity profiles across the center of double slits. The slit width and the spacing between double slits are both 0.4 mm. *c*, Super-resolution image of a coiled slit by using the meta-lens. *d*, Conventional image of the coiled slit by using the periodic-belt Fresnel zone plate. *e*, Super-resolution image of a hole array by using the meta-lens. *f*, Conventional image of the hole array by using the periodic-belt Fresnel zone plate.

II. Revisions in Supplemental Material

1. In lines 34-47, page 4, revised figure and caption

Figure S4 / *The imaging comparison of different lenses. a, The fabricated samples of a single-belt lens (left), a periodic-belt lens (middle) and a super-oscillation meta-lens (right). b, Ultrasound images of a coiled slit. Left: conventional imaging via a single-belt Fresnel zone plate. Middle: conventional imaging via a periodic-belt Fresnel zone plate. Right: super-resolution imaging via a meta-lens. c, Ultrasound images of a hole array. Left: conventional imaging via a single-belt Fresnel zone plate. Middle: conventional imaging via a periodic-belt Fresnel zone plate. Right: super-resolution imaging via a meta-lens. d, The intensity distributions along the dashed lines in b. e, The intensity distributions along the dashed lines in c. For the single-belt Fresnel zone plate, the radius $R_n = 7.5$ mm the width $\Delta r = 0.75$ mm the imaging plane locates at $z = 5.2\lambda$ with the focused sound intensity satisfying*

$I = C_n |J_0(kr \sin \alpha_n)|^2$. For the periodic-belt Fresnel zone plate, the period $p = 1.65$ mm the width $\Delta r = 0.75$ mm the imaging plane locates at $z = 5.2\lambda$.

2. In lines 48-64, pages 5-6, added figures and captions

Figure S5 / The deconvolution in pressure field post-processing. In experimental measurements, the measured field distribution is the solution function $g_{G_r \times G_c}$. The aperture function of hydrophone is the convolutional interaction function $h_{H_r \times H_c}$. The real pressure field distribution is the source function $f_{F_r \times F_c}$.

Figure S6 / The deconvolution in deciphering the super-oscillation field. Left: the post-processed intensity field of the super-oscillation packet. Middle: the aperture

function of hydrophone. Right: the measured intensity field in experiments, which is actually the convolution between the post-processed field and the aperture function.

Figure S7 | The intensity curves and local wavenumbers. a, The normalized intensity distributions of a super-oscillatory field (black line) and the maximum spatial frequency component (red line). The bottom arrows indicate the location of the diffraction limit (0.38λ). **b**, The calculated local wavenumbers of a super-oscillatory field (black line) and the maximum spatial frequency component (red line).

3. In lines 69-87, page 7, added text

Supplementary Note 1 | Convolution and deconvolution

In mathematics, the convolution between the functions $f(x,y)$ and $h(x,y)$ is defined by

$$\begin{aligned} g(x,y) &= f(x,y) * h(x,y) \\ &= \iint f(\xi,\eta)h(x-\xi,y-\eta)d\xi d\eta, \end{aligned} \quad (S1)$$

where $*$ denotes the convolution operator, $f(x,y)$ is the source function before the convolution process, $h(x,y)$ is the convolutional interaction function and $g(x,y)$ is the solution function after convolution. In experimental measurements, the

functions $f(x, y)$, $h(x, y)$ and $g(x, y)$ are discretized into matrices. Then the Eq.

(S1) can be expressed into

$$g(s, t) = \sum_{m=0}^{F_r-1} \sum_{n=0}^{F_c-1} f(m, n)h(s-m, t-n), \quad (S2)$$

where (m, n) and (s, t) denote the element indices of the matrices $f_{F_r \times F_c}$ and $h_{H_r \times H_c}$. F_r and F_c represent the numbers of rows and columns of the source function matrix $f_{F_r \times F_c}$. The pressure field scanning is actually a convolution process, where the measured field distribution can be regarded as the solution function $g(x, y)$. The aperture function of hydrophone is the convolutional interaction function $h(x, y)$, which is a truncated Gaussian function with the FWHM ~ 2 mm (size of the truncated region: ~ 0.6 mm). The source function $f(x, y)$ is the one that reflects the real pressure field distribution, which can be solved by the deconvolution process, as schematically described in Figs. S5 and S6.

4. In lines 224-229, pages 14-15, revised text

From Eq. (S25), the acoustic radiation force \mathbf{F}^{rad} is expressed in terms of the velocity potential of incident acoustic wave ϕ_m at the particle position as well as the scattering coefficients of the monopole and dipole components, namely, f_1 and f_2 .

Based on the previous study, the monopole scattering coefficient $f_1 = 1 - \frac{\rho_0 c_0^2}{\rho_p c_p^2}$ and

dipole scattering coefficient $f_2 = \frac{2(\rho_p - \rho_0)}{2\rho_p + \rho_0}$, where c_0 and ρ_0 are the sound speed

and the density of water; c_p and ρ_p denote the sound speed and the density of tiny particles.

Reviewers' Comments:

Reviewer #1:

Remarks to the Author:

The authors have fully considered my previous comments, addressed all the questions and revised the manuscript accordingly. Deconvolution has been made in Fig. 3e and Fig. 4b which give better agreement with the simulation results, although the first experimental minimum doesn't get close to zero due to the practical limitation in the experiment. More importantly, the authors have done the supplemental experiment to convince the resolution improvement using the super-oscillatory meta-lens by a fair comparison with a Fresnel zone plate with a similar size and numerical aperture.

I can recommend this work to be accepted for publication on Nature Communications and hope it will inspire more researchers in the acoustic community to get to know the super-oscillation phenomenon and find more interesting imaging applications therein.

Reviewed by Yuan Guanghui

Reviewer #2:

Remarks to the Author:

The points raised in the previous review have been satisfactorily addressed and I agree that this work can be accepted for publication on Nature Communications.

Response to the reviewer's comments

Reviewer #1 (Remarks to the Author):

The authors have fully considered my previous comments, addressed all the questions and revised the manuscript accordingly. Deconvolution has been made in Fig. 3e and Fig. 4b which give better agreement with the simulation results, although the first experimental minimum doesn't get close to zero due to the practical limitation in the experiment. More importantly, the authors have done the supplemental experiment to convince the resolution improvement using the super-oscillatory meta-lens by a fair comparison with a Fresnel zone plate with a similar size and numerical aperture.

I can recommend this work to be accepted for publication on Nature Communications and hope it will inspire more researchers in the acoustic community to get to know the super-oscillation phenomenon and find more interesting imaging applications therein.

Reviewed by Yuan Guanghui

Reply: Great thanks for the reviewer's positive comments.

Reviewer #2 (Remarks to the Author):

The points raised in the previous review have been satisfactorily addressed and I agree that this work can be accepted for publication on Nature Communications.

Reply: Great thanks for the reviewer's positive comments.